# Detecting structural heterogeneity in single-molecule localization microscopy data

Teun A.P.M. Huijben [1,4], Hamidreza Heydarian [1,4], Alexander Auer [2,3], Florian Schueder [2,3], Ralf Jungmann [2,3], Sjoerd Stallinga [1] & Bernd Rieger [1✉]

Particle fusion for single molecule localization microscopy improves signal-to-noise ratio and overcomes underlabeling, but ignores structural heterogeneity or conformational variability. We present a-priori knowledge-free unsupervised classification of structurally different particles employing the Bhattacharya cost function as dissimilarity metric. We achieve 96% classification accuracy on mixtures of up to four different DNA-origami structures, detect rare classes of origami occuring at 2% rate, and capture variation in ellipticity of nuclear pore complexes.

[1] Department of Imaging Physics, Delft University of Technology, Delft, The Netherlands. [2] Faculty of Physics and Center for Nanoscience, Ludwig Maximilian University, Munich, Germany. [3] Max Planck Institute of Biochemistry, Martinsried, Germany. [4] These authors contributed equally: Teun A.P.M. Huijben, Hamidreza Heydarian. ✉email: b.rieger@tudelft.nl

Single-molecule localization microscopy (SMLM) enables imaging below the diffraction limit[1,2]. The image quality can be improved further by fusing hundreds of super-resolution images of identical bio-molecular structures, further referred to as particles, into a single reconstruction[3–6]. This approach overcomes the problem of incomplete labeling using the central assumption that all particles represent the same underlying structure. In reality, however, the sample might be heterogeneous in structure due to the biology itself[4], sample preparation[7], diseases or drug-induced variations. These potential variations between structures blur standard fusion and small subsets of structurally different particles remain undetected. Sporadic 9-fold symmetric nuclear pores[8,9], for example, cannot be detected in the reconstruction.

Image classification is commonly used in single-particle averaging (SPA) for cryo-electron microscopy (cryo-EM)[10,11] to find the viewing direction of each particle from its projection. Using EM classification techniques as such to SMLM data[6,12] does not employ the full potential of SMLM data, due to the different image formation process, in particular the incomplete labeling[13,14], and the use of localization coordinates instead of pixelated images. Previous work to separate classes in SMLM uses a deep neural network for classification[15]. Here, however, the different classes need to be known a-priori and imaged in separate experiments to form learning sets in order to train the neural network. This strong a-priori knowledge is not compatible with discovering unknown data variation and is, therefore, inapplicable to most cellular imaging applications.

Here, we present an unsupervised classification tool to cluster 2D/3D SMLM data into (rare) structural subclasses without assuming any prior knowledge about the different classes. The method is capable of detecting multiple repeated structures in one field of view or capture structural variations, which is expected for many biological structures due to biological dynamics, deformation of the cellular structure, or phenotypical variations, within one class of particles. We successfully classified experimental DNA-PAINT and (d)STORM datasets, containing either multiple structures, rare subclasses or continuous structural variation. On experimental data, our approach shows 96% classification performance on multi-class DNA-origami data, detects rare classes of mirrored origami structures occurring at a 2% rate, identifies elliptical shape variation in nuclear pore complexes, and reveals height variation in 3D origami tetrahedron structures.

## Results

**Clustering pipeline.** Our approach uses pairwise registration of all particles to obtain a dissimilarity metric. Subsequently, a feature space is obtained by performing multidimensional scaling (MDS)[16] on the dissimilarity representation which is followed by k-means clustering[17] and particle fusion per cluster (Fig. 1). In order to compare every pair of particles, we employ the Bhattacharya cost function, which we used earlier in the all-to-all

registration of template-free particle fusion[5] (Methods). In contrast to this earlier use, we use the optimum value of this cost function as similarity metric, we are not interested in their relative translations and rotations. The Bhattacharya metric works directly on the localization data, takes (possibly anisotropic) localization uncertainties into account, and is robust against underlabelling. The pairwise registration of $N$ particles results in $N(N-1)/2$ similarity values. After converting the similarity to dissimilarity values, we use MDS to translate the pairwise dissimilarities into spatial coordinates of the particles in a multi-dimensional space[16]. This constellation preserves the pairwise dissimilarities by minimizing the so-called metric stress loss function (Methods).

We used k-means clustering in the multidimensional scaling space[17] in order to group the particles into clusters. K-means clustering has several advantages over other existing approaches that can be used on the MDS maps. First of all, it has the best performance on spherical clusters which is the dominant cluster shape appearing in our MDS space (Supplementary Fig. 1). Secondly, it is fast (compared to for example mean-shift) and has only one tuning parameter, $K$. Mean-shift clustering, which also only has one free parameter, can potentially be used as an alternative to k-means, however, the bandwidth size selection itself is challenging and in our case does not have a physical meaning as the distance between the points in the MDS maps is based on the dissimilarity values. Other complex clustering approaches like DBSCAN or OPTICS[18] have similar difficulties in tuning even more free parameters which are prohibitive for non-expert end-users.

In case of the absence of prior knowledge about the number of subclasses present in the data, we propose a two-step approach for determining the optimal number of clusters ($K$) for the k-means algorithm. Firstly, the user can inspect the silhouette cluster evaluation values[19] (see Methods), for a range of $K$ and choose for the one which gives the highest average score for all particles (Supplementary Fig. 2). Secondly, the user can inspect the scatter plot of the first three dimensions of the MDS space to determine the proper value of $K$ (Supplementary Fig. 1). In certain situations, such as highly imbalanced subclasses, the silhouette graph would not have a predominant peak and visual inspection of the MDS maps may not be helpful to infer the number of underlying classes. Here, the general advice is to choose a large value of $K$. This will result in many small classes which can subsequently be grouped further using the eigen image approach (Methods). Reconstruction per cluster is quick since the computationally most intensive procedure, the all-to-all registration, is already performed prior to the classification. This fast reconstruction makes it possible to investigate multiple values for the number of clusters, $K$.

**Clustering of multi-class SMLM data imaged separately.** We applied our classification algorithm to different multi-class

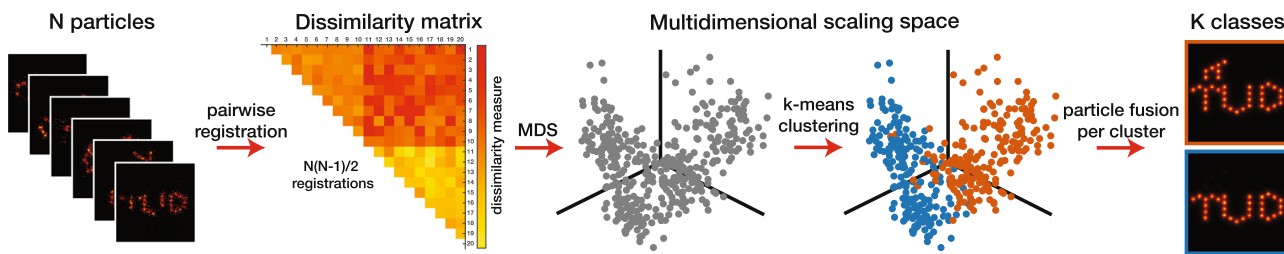

**Fig. 1 Classification pipeline.** $N$ particles are pairwise registered, resulting in $N(N-1)/2$ dissimilarity values. Multidimensional scaling (MDS) embeds the elements of the dissimilarity matrix in a multidimensional space (only the first 3 dimensions are shown). K-means clustering in this space results in $K$ clusters and the particles are fused per cluster.

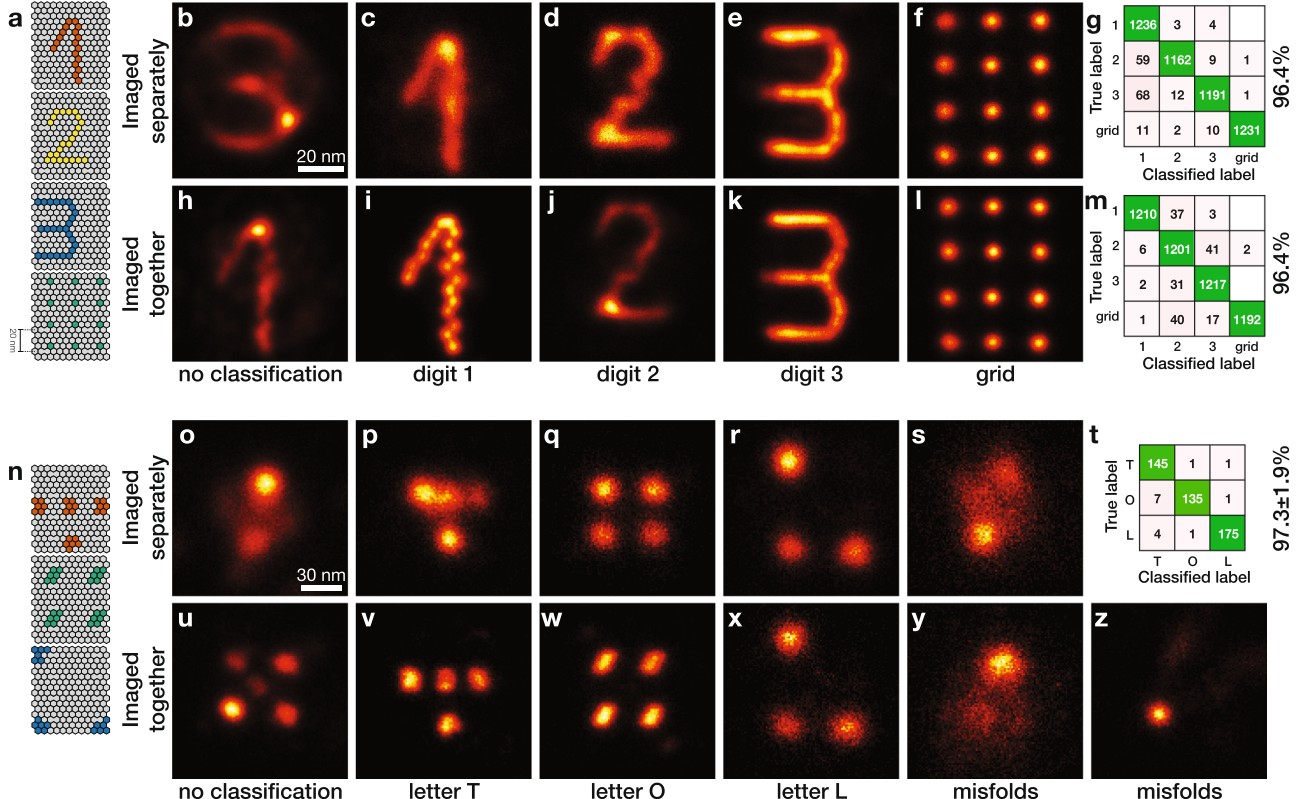

**Fig. 2 Classification of experimental DNA origami datasets. a** Templates of four DNA-origami designs in the digits dataset: the digits 1, 2, and 3 and a 20 nm grid. **b** Fusion result without classification of 2000 particles (randomly selected from a set of 10,000 images, 2500 per class). This data is imaged separately per class and combined into one dataset prior to the classification. **c–f** The four classes resulting from the classification of 5000 images (randomly selected from a set of 10,000 images, 2500 per class) containing 1374, 1179, 1214, and 1233 particles, respectively. **g** Confusion matrix of the classifications **c–f** with an overall performance of 96.4%. **h** Fusion result without classification of 2000 particles imaged in one FOV. **i–l** The four classes resulting from the classification of 5000 particles images in one FOV, containing 1219, 1309, 1278, and 1194 particles, respectively. **m** Confusion matrix of the classifications **i–l**, with an overall performance of 96.4%. **n** Templates of three DNA-origami designs in the letters dataset: letters T, O, and L. **o** Fusion result without classification of 600 particles (200 per class). This data is imaged separately per class and combined into one dataset prior to the classification. **p–s** The four classes resulting from the classification of **o**, containing 207, 122, 176, and 95 particles, respectively. **t** Average confusion matrix of two independent classifications performed as in **p–s** with an average performance of 97.3 ± 1.9%, where the class of misfolds, **s**, is not taken into account. **u** Fusion result without classification of 800 particles imaged in one FOV. **v–z** The five classes resulting from the classification of **u**, containing 170, 238, 130, 139, and 123 particles, respectively. Scale bar of **b** applies to **c–f** and **h–l**. Scale bar of **o** applies to **p–s** and **u–z**.

SMLM datasets in Figs. 2 and 3. We show the nanoTRON dataset[15] (further referred to as digits), which consisted of four classes: the digits 1, 2, 3, and a 3 × 4 grid (Fig. 2a). Each class comprises 2500 instances of each structure. The data were imaged separately per class. We randomly selected 5000 particles from all classes and fused them without classification. This resulted in a blurred reconstruction which resembled vaguely the digit 3 (Fig. 2b), caused by the different numbers of localizations per class (Supplementary Fig. 3a). Classification of the particles followed by fusion, however, resulted in four clearly distinct classes (Fig. 2c–f). The classification performance, which we define as the percentage of correctly classified particles, is 96.4%, and can be calculated since the ground truth class of each particle is known. The confusion matrix (Fig. 2g) gives further insight into the performance by indicating the false/true positives/negatives per class.

**Clustering of multi-class SMLM data imaged together.** To demonstrate that the classification is not selecting on different imaging conditions between the four classes, we repeated the same classification procedure on 5000 particles, 1250 per class, where all four designs were mixed in a single sample and acquired together. Here, the fusion without classification showed the digit

1, again explained by different numbers of localizations (Supplementary Fig. 3b). The fusions after classification in Fig. 2h–l show similar improvements as before, with an overall classification performance of 96.4% (Fig. 2m), using manual labeling[15] as ground truth. The worse reconstruction of digit 1 (Fig. 2c) and digit 2 (Fig. 2j) compared to other classes, is due to lower quality of the individual particles in this experiment (Supplementary Fig. 4), not to suboptimal classification performance. The Fourier ring correlation (FRC) image resolution[20] after reconstruction ranges from 3.7 nm to 5.7 nm per class (Supplementary Fig. 5).

**Clustering of multi-class SMLM data with misfold class.** We further tested our algorithm on a more challenging DNA-origami dataset imaged with DNA-PAINT (Methods), containing three designed structures: the letters T, O, and L (compare Fig. 2n). These structures are harder to separate by classification, due to the presence of a significant number of misfolded structures in the dataset (Supplementary Fig. 6) and a large variation in localization uncertainty between the classes (Supplementary Fig. 7m). We investigated a set containing 600 particles (200 per class) which were imaged separately. Reconstruction without classification gave an unrecognizable outcome (Fig. 2o), whereas classification prior to fusion using $K = 3$ (as suggested by the MDS

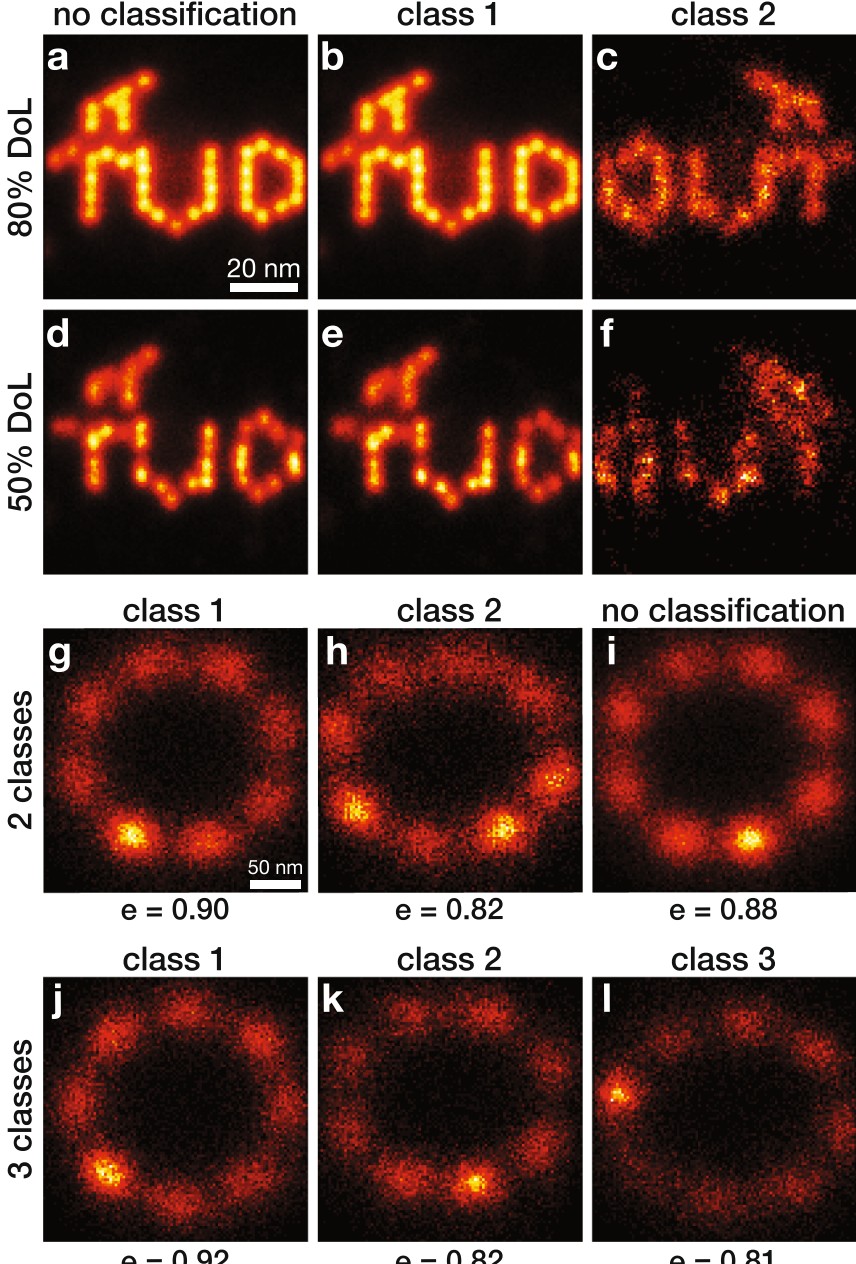

**Fig. 3 Classification of flipped DNA Origami data and NPC data. a** Fusion result without classification of 456 DNA-origami structures of the TUD-logo with 80% DoL. **b**, **c** The two classes resulting from the classification of **a**, containing 446 (normal orientation) and 10 (flipped orientation) particles per class, respectively. **d–f** Same as **a–c**, but with 50% DoL and 381 and 8 particles per class, respectively. **g**, **h** The two classes resulting from the classification of **i**, containing 168 and 136 particles, respectively. We used $K = 4$ for the 80% DoL data and $K = 40$ for the 50% DoL data, followed by further grouping with the eigen image approach with $C = 2$ (Methods). **i** Fusion result without classification of 304 NPCs. **j–l** Same as **g**, **h**, but with classification into three classes, with 133, 90 and 81 particles per class, respectively. For **g–l**, the ellipticity values $e$ are defined as the ratio of the major axis over the minor axis of the ellipse fitted to the localizations, and are indicated below each class. Scale bar of **a** applies to **b–f**. Scale bar of **g** applies to **h–l**.

scatter plots in Supplementary Fig. 1d–f and the silhouette plot in Supplementary Fig. 2c) resulted in clearly visible classes (Supplementary Fig. 6a–c) and a classification performance of 89% (Supplementary Fig. 6d). In contrast to the digits dataset, this dataset contained misfolded and unrecognizable particles (Supplementary Fig. 6h), because the simple design of only a few dots makes it hard for particle picking to exclude misfolded structures. Therefore, classification into four classes (Fig. 2p-s) was needed to capture the misfolded structures into a separate class (Fig. 2s). An average classification performance of 97.3 ± 1.9% was obtained

for two independent repeats (Fig. 2t). The same three classes were visible after classification when all three designs were acquired from a single sample (Fig. 2u–w). Again, $K = 3$ (suggested by the analysis of MDS scatter plots and silhouette plots) was sufficient to correctly classify the structures (Supplementary Fig. 7a–c), but two additional misfold classes were needed to get correct reconstructions as the misfolds themselves vary (Supplementary Fig. 7). The digits dataset did not need a separate misfold class, as particle picking was successful in automatically removing all misfolded particles.

**Detection of rare subclasses**. In addition to datasets that contain multiple, equally abundant classes, we tested the classification algorithm on experimental data with skewed distributions of the number of particles per class. The TUD-logo DNA-origami data[5] contains a small fraction that is unintentionally imaged upside-down, resulting in mirrored logos. Normally, these mirrored images are manually removed from data before fusion. Both for 80% and 50% density of labeling (DoL), the mirrored particles were not visible when all of them are fused (Fig. 3a, d), due to their low abundance. However, classification found these classes containing, respectively, 10 and 8 mirrored particles (Fig. 3c, f), from a total of 456 and 381 particles. We used $K = 4$ for the 80% DoL data (guided by visual inspection of the MDS scatter plot in Supplementary Fig. 1g–i) and $K = 40$ and for the 50% DoL data, both followed by further grouping (Methods) into two final classes. For the 50% DoL data, neither the MDS scatter plots nor the silhouette plots (Supplementary Fig. 2f) could help in determining the optimal value for $K$. Following our rule of thumb to use a high value and investigate the outcome for multiple values, $K = 40$ appeared to work best. To test how low we can go in the detection of rare events, we also applied our method to simulated nuclear pore complexes (NPC), with a rare 9-fold symmetric class. Classification could find these rare cases, even when only 2% was 9-fold symmetric (Supplementary Fig. 8). For rare class detection, it is necessary to have a high total number of particles as the rare class needs at least approximately 10 particles for successful detection.

**Detection of structural variations in 2D SMLM data**. Next to DNA-PAINT data, we applied our method to data[3] acquired with (d)STORM of the integral membrane protein gp210, part of the NPC. Previously, Heydarian et al.[5] used prior knowledge about the 8-fold symmetry for particle fusion and obtained a circular reconstruction with a homogeneous distribution of localizations. We applied our method to the same data and found classes with a range of ellipses (Fig. 3g–l). The classification separated the particles on their ellipticity (Supplementary Fig. 9c, d). The major axes of the ellipses align in the field-of-view (Supplementary Fig. 9a, b), suggesting that the deformations were caused either by the sample preparation or by the projection of the tilted rings onto the image plane. Since a tilt of 25° is needed to create the detected elliptical deformation, we think that the sample preparation is the more likely root cause for the found ellipticity. We further characterized and quantified the classification performance of our method on simulated data with discrete variation, TUD-logos with and without flame, and continuous variation, NPCs with a varying diameter (Supplementary Fig. 10). In the latter case, the classification perfectly separates the NPCs on diameter.

**Detection of structural variations in 3D SMLM data**. Finally, we applied our classification pipeline to PAINT images of a tetrahedron-shaped 3D DNA-Origami nanostructures (Methods) with an edge length of ~100 nm (height ~ 90 nm). Our analysis shows that there is a variation in the height distribution of the tetrahedrons. This is consistent with the fact that the structural stability of large DNA origami nanostructures is limited. Before classification (Supplementary Fig. 11a), the histogram of the z-coordinate of the localization data suggests a height of 90 nm. However, the distribution of the localizations around the binding sites are clearly elongated along the optical axis larger than the mean axial localization uncertainty of 8 nm. Classification of the tetrahedrons and within-class fusion of them results in a more isotropic resolution of the reconstructions (Supplementary Fig. 11b, c). Here, we identified different heights with a mean

height of 55 and 95 nm or 45, 65, and 95 nm depending on the number of quantization levels of the continuous height distribution. The data suggest a continuous distribution of heights, which by definition does not provide the user with a preferred number of clusters, allowing the user to divide the distribution into a desired number of clusters. This analysis shows that the elongation of the binding sites along the z-axis in the reconstruction of all 218 tetrahedrons is due to the structural heterogeneity of the DNA-origami structures and not due to the poor imaging conditions.

## Discussion

The developed classification pipeline (Fig. 1) is based on the dissimilarities between all the particles. The clustering is performed by using MDS as a spatial embedding of the particles, followed by k-means clustering. An alternative would be to directly cluster based on the dissimilarity values using hierarchical clustering (HAC). We have implemented and optimized both strategies and tested them on multiple datasets. The performance of the first method, i.e., clustering based on a spatial embedding, performs significantly better than hierarchical clustering (Supplementary Note 1, Supplementary Fig. 12). Therefore, the first method, is used for all classifications in this paper. Sabinina et al.[21] recently proposed a similar HAC approach, however, with a different dissimilarity metric. Easy-to-classify data can be classified correctly by both approaches, although HAC does require the delicate choice of the right parameters, like the distance criterion and the pruning threshold. Other advantages of our approach are that there is no choice for the linkage criterion and that no particles are lost in pruning the dendrogram, because of the creation of single-particle clusters. We can conclude that our MDS approach performs significantly better than HAC and has fewer tuning parameters, which supports the decision to use the MDS approach in this paper.

In summary, we have developed an a-priori knowledge-free classification tool to identify (rare) structural subclasses in SMLM data. Once data heterogeneity is found, particle fusion algorithms can be employed. We successfully classified experimental DNA-PAINT and (d)STORM datasets, containing either multiple structures, rare subclasses or continuous structural variation.

## Methods

**DNA origami design and self-assembly of letters dataset**. The DNA origami structures were designed with the software package *Picasso*[22] using the *Design* module. The DNA origami were synthesized in a one-pot reaction with 50 µl total volume containing 10 nM scaffold strand, 100 nM core staples, 1 µM biotinylated staples and 1 µM DNA-PAINT docking sites staples. The DNA sequences are listed in Supplementary Tables 1–4. The folding buffer was 1× TE buffer with 12.5 mM MgCl$_2$. The DNA origami were annealed using a thermal ramp. First, incubating for 5 min at 80 °C, then going from 65 °C to 4 °C over the course of 3 h. DNA origami structures were purified via two rounds of PEG precipitation by adding the same volume of PEG-buffer, centrifuging at 14,000 × g at 4 °C for 30 min, removing the supernatant and resuspending in folding buffer. The 2× PEG buffer stock was 1× TE buffer with 500 mM NaCl and 15% PEG-8000 (v/w).

**Optical setup for the acquisition of letters dataset**. Fluorescence imaging was carried out on an inverted microscope (Nikon Instruments, Eclipse Ti2) with the Perfect Focus System, applying a custom-built objective-type TIRF configuration with an oil-immersion objective (Nikon Instruments, SR HP Apo TIRF 100XC, numerical aperture 1.49, Oil). A 560 nm and a 642 nm laser (both MPB Communications Inc., 500 mW, DPSS-system) were used for excitation. The coaxially aligned laser beams were passed through a quad-band cleanup filter (Chroma Technology, ZET405/488/561/640xv2) and coupled into the microscope objective using a quad-band beam splitter (Chroma Technology, ZT405/488/561/640rpcv2). Fluorescence light was spectrally filtered with a quad-band emission filter (Chroma Technology, ZET405/488/561/640m-TRFv2) and imaged on a sCMOS camera (Andor, Zyla 4.2 Plus) without further magnification, resulting in an effective pixel size of 130 nm (after 2 × 2 binning).

**DNA-PAINT sample preparation of letter dataset**. For chamber preparation, a piece of coverslip (no. 1.5, 18 × 18 mm, ~0.17 mm thick) and a glass slide (76 × 26 mm, 1 mm thick) were sandwiched together by two strips of double-sided tape to form a flow chamber with inner volume of ~20 μl. First, 20 μl of biotin-labeled bovine albumin (1 mg/ml, dissolved in buffer A) was flown into the chamber and incubated for 2 min. Then the chamber was washed using 40 μl of buffer A. Second, 20 μl of streptavidin (0.5 mg/ml, dissolved in buffer A) was then flown through the chamber and incubated for 2 min. Next, the chamber was washed with 20 μl of buffer A and subsequently with 20 μl of buffer B. Then ~500 pM of the DNA origami structures were flown into the chamber and allowed to attach to the surface for 2 min. Finally, the imaging buffer with buffer B with dye-labeled imager strands was flowed into the chamber and sealed with silicon. Imager sequences are stated in Supplementary Table 5. Imaging conditions are listed in Supplementary Tables 6–9.

Two buffers were used for sample preparation and imaging: buffer A (10 mM Tris-HCl pH 7.5, 100 mM NaCl, 0.05% Tween 20, pH 7.5); buffer B (5 mM Tris-HCl pH 8, 10 mM MgCl$_2$, 1 mM EDTA, 0.05% Tween 20, pH 8). Imaging Buffer B was supplemented with: 1× Trolox, 1× PCA and 1× PCD. Trolox, PCA and PCD stocks were as follows: 100× Trolox: 100 mg Trolox, 430 μl 100% methanol, 345 μl 1 M NaOH in 3.2 ml H2O. 40× PCA: 154 mg PCA, 10 ml water and NaOH were mixed and the pH was adjusted to 9.0. 100× PCD: 9.3 mg PCD, 13.3 ml of buffer was used (100 mM Tris-HCl pH 8, 50 mM KCl, 1 mM EDTA, 50% glycerol).

**DNA-origami design and self-assembly of tetrahedron dataset**. The tetrahedron DNA-origami structures were formed in a one-pot reaction with a 50 μl total volume containing 10 nM scaffold strand (p8064), 100 nM core staples, 100 nM connector staples, 100 nM vertex staples, 100 nM biotin handles, 100 nM DNA-PAINT docking site staples, and 1400 nM biotin anti-handles in folding buffer (1× TE (5 mM Tris, 1 mM EDTA) buffer with 10 mM MgCl2). The solution was annealed using a thermal ramp cooling from 80 to 4 °C over the course of 15 h. After self-assembly, the structures were mixed with 1× loading dye and then purified by agarose gel electrophoresis (1.5% agarose, 0.5× TAE, 10 mM MgCl2, 1× SYBR Safe) at 3 V/cm for 3 h. Gel bands were cut, crushed and filled into a Freeze 'N Squeeze column and spun for 5 min at 1000 × g at 4 °C.

**Optical setup for the acquisition of tetrahedron dataset**. Fluorescence imaging was carried out on an inverted microscope (Nikon Instruments, Eclipse Ti) with the Perfect Focus System, applying an objective-type TIRF configuration with an oil-immersion objective (Nikon Instruments, Apo SR TIRF ×100, numerical aperture 1.49, Oil). A 561 nm (200 mW, Coherent Sapphire) laser was used for excitation. The laser beam was passed through cleanup filters (Chroma Technology, ZET561/10) and coupled into the microscope objective using a beam splitter (Chroma Technology, ZT561rdc). Fluorescence light was spectrally filtered with an emission filter (Chroma Technology, ET600/50m, and ET575lp) and imaged on an sCMOS camera (Andor, Zyla 4.2 Plus) without further magnification, resulting in an effective pixel size of 130 nm (after 2 × 2 binning).

**DNA-PAINT sample preparation of tetrahedron structures**. For chamber preparation, a piece of coverslip (no. 1.5, 18 × 18 mm2, ~0.17 mm thickness) and a glass slide (3 × 1 inch2 1 mm thick) were sandwiched together by two strips of double-sided tape to form a flow chamber with inner volume of ~20 μl. First, 20 μl of biotin-labeled bovine albumin (1 mg/ml, dissolved in buffer A) was flown into the chamber and incubated for 2 min. Then the chamber was washed using 40 μl of buffer A. Second, 20 μl of streptavidin (0.5 mg/ml, dissolved in buffer A) was then flown through the chamber and incubated for 2 min. Next, the chamber was washed with 40 μl of buffer A and subsequently with 40 μl of buffer B. Then ~50 pM of the tetrahedron DNA origami structures were flown into the chamber and allowed to bind for 30 min. Afterward the chamber was washed with 40 μl of buffer B again. Finally, the imaging buffer with buffer B and 1× Trolox, 1× PCA, and 1× PCD with the Cy3b-labeled imager strand (P1, 9nt at 3 nM) was flown into the chamber. The chamber was sealed with epoxy before subsequent imaging.

Two buffers were used for sample preparation and imaging: buffer A (10 mM Tris-HCl pH 7.5, 100 mM NaCl, 0.05% Tween 20, pH 7.5); buffer B (10 mM MgCl2, 5 mM Tris-HCl pH 8, 1 mM EDTA, 0.05% Tween 20, pH 8). The imaging buffer was supplemented with: 1× Trolox, 1× PCA and 1× PCD Trolox, PCA and PCD were as follows: 100× Trolox:100 mg Trolox, 430 μl 100% methanol, 345 μl 1 M NaOH in 3.2 ml H2O. 40× PCA: 154 mg PCA, 10 ml water and NaOH were mixed and the pH was adjusted to 9.0. 100× PCD:9.3 mg PCD, 13.3 ml of buffer was used (100 mM Tris-HCl pH 8, 50 mM KCl, 1 mM EDTA, 50% glycerol).

**DNA-PAINT super-resolution microscopy of tetrahedron dataset**. Images were acquired with an imager strand concentration of 3 nM (P1-Cy3B, 9nt) in imaging buffer. Here, 30,000 frames were acquired at 300 ms exposure time. The readout bandwidth was set to 200 MHz. Laser power (at 561 nm) was set to 110 mW (measured at the BFP of the objective). This power corresponds to an intensity of ~990 W cm-2 at the sample plane.

**Super-resolution reconstruction**. Raw fluorescence data were subjected to spot-finding and subsequent super-resolution reconstruction with the *Picasso* software package using the *Localize* module. The drift correction was performed with

*Picasso Render* using a redundant cross-correlation (RCC) with segmentation set to 1000 frames. The remaining drift was corrected with the *Render*'s function *Undrift from picked* with all picked DNA-origami structures. The DNA-origami were picked using *Render Pick similar*.

**Materials**. Unmodified DNA oligonucleotides, fluorescently modified DNA oligonucleotides, and biotinylated DNA oligonucleotides were purchased from MWG Eurofins. M13mp18 scaffold was obtained from Tilibit. BSA-Biotin was obtained from Sigma-Aldrich (cat: A8549). Streptavidin was ordered from Invitrogen (cat: S-888). Tris 1 M pH 8.0 (cat: AM9856), EDTA 0.5 M pH 8.0 (cat: AM9261), Magnesium 1 M (cat: AM9530G) and Sodium Chloride 5 M (cat: AM9759) were ordered from Ambion. Ultrapure water (cat: 10977-035) was purchased from Gibco. Polyethylene glycol (PEG)-8000 (catalog no. 6510-1KG) was purchased from Merck. Tween-20 (P9416-50ML), glycerol (65516-500 ml), methanol (32213-2.5 L), protocatechuate 3,4-dioxygenase pseudomonas (PCD; P8279), 3,4-dihydroxybenzoic acid (PCA; 37580-25G-F) and (±)-6-hydroxy- 2,5,7,8-tetra-methylchromane-2-carboxylic acid (trolox; 238813-5 G) were ordered from Sigma-Aldrich. Glass slides (cat: 48811-703) were obtained from VWR. Coverslips were purchased from Marienfeld (cat: 0107032). Silicon (cat.1300 1000) was ordered from picodent. Double-sided tape (cat: 665D) was ordered from Scotch.

**Classification**. The developed clustering or classification pipeline (Fig. 1) consists of four main building blocks: (1) pairwise registration of all particles resulting in an upper triangular matrix of dissimilarity values; (2) multidimensional scaling based on the dissimilarity representation; (3) k-means clustering of the multidimensional embedding; and (4) per cluster particle fusion.

**Pairwise registration**. The registration of every particle to every other particle is performed as in Heydarian et al.[5]. A Gaussian-mixture-model-based (GMM) registration in combination with the Bhattacharya cost function results in transformation parameters and a cost function value for each pair of particles. The GMM registration[23] finds the optimal transformation by placing a Gaussian distribution onto every localization and maximizing the overall overlap between all Gaussians of the two particles. The Gaussian distributions are all given the same width, the so-called scale, which is a parameter that is dataset specific. The optimal scale parameter is determined by performing a scale-sweep in the range 0.001–0.5 camera pixels (corresponding to 0.13–65 nm). Since the minimum spacing between fluorophore binding sites ranges from 5 nm (DNA origami) to 60 nm (Xenopus nuclear pore), this range is sufficient. A smaller scale would result in overfitting on individual localizations and a larger scale would blur the localizations belonging to different binding sites. In the scale-sweep, 10 random combinations of particles are registered using GMM for 50 scales linearly distributed over the above range. The optimal scale parameter corresponds to the maximum GMM value after normalizing and averaging the obtained 10 GMM values over all scales. This procedure results in a scale value of 0.03 for the digits, 0.15 for the letters, 0.1 for NPC and 0.01 for the TUD-logo (in camera pixels). Each GMM registration is initialized with six angles (uniformly distributed between 0 and $2\pi$) and the optimal registration is determined by evaluating the Bhattacharya cost function (Eq. (1) of Heydarian et al.[5]) on the six found transformations. The Bhattacharya cost function is adapted by adding a normalization with respect to the number of localizations and with respect to the localization uncertainties as follows:

$$S(a, b) = \frac{1}{K_a K_b} \sum_{i=1}^{K_a} \sum_{j=1}^{K_b} \frac{1}{(\sigma_{a,i}^2 + \sigma_{b,j}^2)} \exp\left(-\frac{1}{2} \frac{(\boldsymbol{r}_{a,i} - \boldsymbol{r}_{b,j})^2}{\sigma_{a,i}^2 + \sigma_{b,j}^2}\right), \quad (1)$$

where $K_a$ and $K_b$ are the number of localizations, $\boldsymbol{r}_a$ and $\boldsymbol{r}_b$ the localization coordinates and $\sigma_a$ and $\sigma_b$ the isotropic localization uncertainties for particles $a$ and $b$, respectively. Both normalizations are crucial to reliably compare the similarity of different pairs of particles. Without the normalizing with respect to the number of localizations, particles with a high number of localizations will give a high-cost function value. This is unfavorable, since we want the cost function to reflect the degree of similarity between particles, not the number of localizations. The normalization with respect to the uncertainty is necessary to prevent that localizations with a high uncertainty result in a high-cost function value. The cost function essentially calculates the overlap between two Gaussian mixtures. Without normalization, the area under the curve is not equal to 1, and the overlap can become very high for large uncertainties.

The pairwise registration of $N$ particles results in an upper triangular matrix of $N(N-1)/2$ cost function values, where a higher value indicates a better match between the particles. The cost function values $S$ are converted to dissimilarity values $D$ by subtracting all of them from the highest value in the matrix:

$$D(a, b) = \max(S) - S(a, b). \quad (2)$$

**Multidimensional scaling**. To cluster the particles based on the dissimilarity matrix, multidimensional scaling (MDS)[16] is used to embed the particles in a multidimensional space. MDS provides a spatial representation of the data by a nonlinear mapping while preserving the pairwise, symmetric dissimilarities between the particles. Essentially, every particle is given a position in space, in such

a way, that the Euclidean distances between pairs of particles are approximately equal to their dissimilarity measure. In this way, particles with a high dissimilarity will be placed far apart in the new space, and particles with a low dissimilarity will end up close to each other. Basically, this is the inverse process to determining distances. Instead of having a spatial embedding and measuring the distances between all pairs of particles, we start with the distances (dissimilarities) and try to find their spatial embedding. We use nonclassical MDS which iteratively updates the MDS coordinates by minimizing the metric stress loss function, defined as:

$$\text{Stress} = \left( \frac{\sum_{ij}(d_{ij} - \|x_i - x_j\|)^2}{\sum_{ij}d_{ij}^2} \right)^{1/2}, \tag{3}$$

where $d_{ij}$ is the dissimilarity between particles $i$ and $j$, and $x_i$ the MDS coordinates of particle $i$. This stress loss function assures that the dissimilarities are preserved as the inter-particle distances in the multidimensional space. We choose to embed the particles into 30 dimensions, since we empirically determined that any value between above 15 is sufficient (Supplementary Fig. 13) and we take as rule of thumb twice this value.

**K-means clustering**. K-means clustering in the multidimensional scaling space results in $K$ clusters of particles. To prevent finding suboptimal clusters due to the random initialization of the k-means algorithm, the clustering procedure is repeated 1000 times. Using a high number of repeats is especially important in cases where a group containing a small number of particles has to be found. Here, we advise taking the number of repeats equal to the number of total particles, to assure that every particle is on average at least once selected as initial seed for the k-means algorithm. From all initializations, the clustering with the lowest within-cluster sums of points-to-centroid distances is chosen. To determine the value of $K$, the scatter plot containing the first 3 dimensions of the MDS space can be inspected visually (Supplementary Fig. 1). Reviewing only the first three dimensions is sufficient, since they contain the most important variation within the MDS space and reviewing the order in a higher-dimensional space is visually challenging. Additionally, determining the optimal number of clusters can be guided by using the silhouette measure as cluster evaluation method (Supplementary Fig. 2). If the first dimensions show separated clusters, or the silhouette plot shows a clear peak, $K$ can be chosen to match the number of clusters. However, when the variation between the different classes of particles is small, the MDS scatter plot will be a single point cloud, and does not give a clear indication for the choice of $K$. In this case, as always, selecting the optimal $K$ requires some tweaking. The user can start with an educated guess of $K$ and then vary $K$ to manually inspect what number of clusters is preferred. This can be done quickly as the computationally most time-consuming part of the method is the pairwise registration, which only has to be done once. The fusing of clusters is significantly faster and can, therefore, be repeated for multiple values of $K$. If the goal is to find a small subgroup with an estimated occurrence of 2%, for example, it is advised to use a high value for $K$ (as for the mirrored TUD-logo's with 50% DoL, Fig. 3d–f) or when the variation is a continuous spectrum, a low value for $K$ in the range of 2-3 will suffice (as for the elliptical NPCs, Fig. 3g–l and 3D tetrahedrons Supplementary Fig. 11).

**Particle fusion per cluster**. Each of the clusters found in the previous step is reconstructed according to the particle fusion pipeline of Heydarian et al.[5]. The reconstruction of the grid structure in the digits dataset is done differently. The data contains 12 binding sites at 20 nm spacing in a $4 \times 3$ pattern. Due to the regular, symmetric nature of this structure in combination with an unbalanced number of localizations per binding site, reconstruction with the above-mentioned algorithm is suboptimal. Therefore, the localizations are first clustered per binding site. This is done by using the density-based clustering for applications with noise (DBSCAN) technique[24], with an epsilon value of 0.03 pixels and 4 as the minimal number of points per cluster. Subsequently, the above-mentioned particle fusion algorithm is applied using the centers of the found clusters. The mean uncertainty of localizations per cluster is used as the uncertainty for each center. Afterward, the centers are replaced by all the original localizations of the particles.

**Further clustering using eigen images**. When classifying small subgroups of structurally different particles, for example low-abundant mirrored DNA-origami (Fig. 3a–f) or 9-fold symmetric NPCs (Supplementary Fig. 8), the multidimensional scaling space has to be clustered into many clusters (high value of $K$, see Methods). Here, the number of obtained $K$ clusters is higher than the desired $C$ classes. The clusters can optionally be grouped further using the eigen image method, which exists of four steps: (1) calculation of the eigen images using principal component analysis (PCA); (2) projection of the $K$ clusters onto the first eigen image; (3) hierarchical agglomerative clustering (HAC) on the obtained weights; and (4) merging of the grouped clusters (Supplementary Fig. 14).

The $K$ reconstructed clusters are optionally grouped further using the eigen image method with the aim to find $C < K$ classes. First, we need to align the $K$ clusters with respect to each other, which is done by using the particle fusion pipeline[5]. Instead of fusing the particles, only the final transformation parameters

are applied to align the different clusters. These clusters are converted to pixelated images by binning the localizations in an $N \times N$ grid, with $N = 400$. This number of pixels results in a pixel size of 0.2 nm for the DNA origami experiments and 0.7 nm for the nuclear pore experiments. As a rule of thumb, the pixel size should be about ¼ of the localization uncertainty in the data[20]. A larger pixel size would result in blurring of the images and thereby loss of structural details, whereas a smaller pixel size will make the images too noisy. Thereafter, the images are normalized to have zero mean and a 2-norm of one. The $K$ images are reshaped into column vectors and concatenated into a matrix $X$ of size $N^2 \times K$, where every column represents an image. The eigen images are computed by applying singular-value decomposition (SVD) to the covariance matrix $M = XX^T$. SVD is a generalized version of eigen decomposition that works for non-diagonalizable and even non-square matrices. The decomposition of $XX^T$ results in:

$$XX^T u = s_u v, \tag{4}$$

where $u$ and $v$ are the left- and right-singular vectors, respectively, with associated singular values $s_u$. Since the matrix $XX^T$ is normal (it commutes with its conjugate transpose), the left- and right-singular vectors are identical, and the SVD algorithm is effectively the same as eigen decomposition. The resulting singular vectors, $u$, represent the eigen images and the associated singular value, $s_u$, indicates how much variation of the data is explained by its respective singular vector. Instead of computing the SVD on the immense covariance matrix $M$ (size $N^2 \times N^2$), applying SVD on $X^TX$ (size $K^2 \times K^2$) is computationally less expensive:

$$X^TX \vec{a} = s_a a. \tag{5}$$

After left-multiplying both sides of the equation with $X$:

$$XX^T Xa = s_a Xa, \tag{6}$$

we see this resembles the decomposition of $XX^T$ as shown before (Eq. (4)) and the sought-after singular vector, $u$, can be calculated as $u = Xa$. By reshaping and normalizing the singular vectors, the eigen images of size $N \times N$ are formed. Since the eigen images are sorted based on their singular values, the first one represents most of the variation between the different images. All $K$ images are projected onto the first eigen image and the resulting weights are clustered into $C$ classes by hierarchical agglomerative clustering with an average-linkage criterion[25]. The value of $C$ represents the final number of classes of the classification. For the low-abundant mirrored DNA-origami structures (Fig. 3a–f), as well as for the 9-fold symmetric nuclear pores (Supplementary Fig. 8) we used $C = 2$. The localizations of the $K$ clusters are fused per class (using the particle fusion pipeline of Heydarian et al.[5]) to form the final $C$ output classes.

**Silhouette cluster evaluation**. The silhouette value is a measure of how similar an object is to its own cluster compared to other clusters. The silhouette value ranges from $-1$ to 1, where a high value indicates a big separation between the clusters mutually. The silhouette value for a point $i$ is defined as $(b_i - a_i)/\max(a_i, b_i)$, where $a_i$ is the average distance of this point to all other points in the same cluster, and $b_i$ the minimum average distance to points in a different cluster, minimized over the other clusters. The silhouette values shown in Supplementary Fig. 2 are based on the first three dimensions of the MDS space and represent the average silhouette value over all particles.

**Fitting ellipses to nuclear pore data**. The ellipticity is defined as the ratio of the length of the major axis over the length of the minor axis of the ellipse that is fitted to the data. To make the elliptical fit less sensitive to the imbalanced distribution of localizations over the 8 binding sites, we fit the ellipse to the medians of the 8 blobs. After grouping the localizations into 8 clusters by k-means clustering, the median position if calculated for all the localizations of each cluster and fitted with an ellipse.

**Order parameter**. The order parameter is calculated for the orientation of the elliptical fits of the experimental nuclear pore data (Supplementary Fig. 9) and is defined as $\langle \cos(2(\theta - \bar{\theta})) \rangle$, where $\theta$ represents the orientation of the ellipse, $\bar{\theta}$ the director which is the average angle of all ellipses, and $\langle \dots \rangle$ the population average over all ellipses of the respective class. By definition, a value for the order parameter of 1 indicates complete alignment of all ellipses and a value of 0 indicates a homogeneous random orientation.

## Data availability

Single molecule localization data for the Digits and Letters DNA-origami nanostructures are publically available via 4TU.Research dataset repository[26] (https://doi.org/10.4121/14074091.v1). Source data are provided with this paper.

## Code availability

The software is available for download under the terms of the Apache2.0 license from Github and 4TU Research Data repository[27] at https://github.com/imphys/smlm_classification2d (https://doi.org/10.4121/14135849.v1).

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

## Acknowledgements
We thank Maarten Joosten for valuable feedback. A.A. acknowledges support from the DFG through the Graduate School of Quantitative Biosciences Munich (QBM). This work was supported by the Dutch Research Council (NWO), VICI grant no. 17046, to B. R. and European Research Council (MolMap, grant no. 680241 to R.J. and Nano@cryo, grant no. 648580 to B.R. and H.H.).

## Author contributions
T.A.P.M.H. and H.H. developed the method. T.A.P.M.H. wrote code, performed simulations and analyzed data. B.R. and S.S. directed the research. A.A., F.S and R.J. designed DNA origami and acquired images. T.A.P.M.H. wrote the paper with support from S.S. and B.R. and all authors commented on the paper.

## Competing interests
The authors declare no competing interests.
