## [Peer Review File · Nature Communications]

REVIEWER COMMENTS

Reviewer #1 (Remarks to the Author):

This manuscript presents a method to classify single-molecule localization microscopy (SMLM) images of small structures. In a manner similar to particle classification and class-averaging in single-particle cryo-electron microscopy, this method can help SMLM resolving heterogeneous subcellular structures to understand their structural organizations. Therefore, it is a useful tool to the community. The manuscript validated the method using simulated and experimental data. It is generally well written. The manuscript can be accepted if the comments below are satisfactorily addressed:

1. Clustering method. Any classification method has two components: a distance metric telling the difference between two particle images, and a cluster analysis method. The distance metric in this manuscript is mostly based on what the authors previously established for single particle image alignment, and the manuscript used the centroid-based K-means method for cluster analysis. While K-means is widely used and has the best performance when the clusters are more or less spherical, there are many other cluster analysis methods available, some of them avoiding the limitation of K-means that a pre-defined cluster number is needed. It is therefore important for the manuscript to:

(1.1) Provide a rationale of why K-mean method is chosen (even a very simple one) and a brief discussion of other cluster analysis methods potentially suitable for this task, as well as their advantages and drawbacks.

(1.2) Establish an objective way of determining the cluster number, K , and an objective metric to test the validity and quality of cluster analysis results, especially for samples without sufficient prior knowledge. Currently, the determination of K is rather arbitrary and highly subjective. For example, in the case of analyzing the TOL origami images, it is already known that three types of structures exist in the data. However, setting $K = 3$ resulted in seemingly acceptable clustering but completely missed the L class. L was observed only when increasing K to 5 (not even 4). In other words, the L class will not be discovered unless its existence is already known and parameters are tweaked specifically for its appearance. This is a huge limitation to practical applications in making biological discoveries. Similarly, the manuscript states that a large K needs to be used for the purpose of discovering rare populations, which is another case where user choices affect analysis outcome. It is understood that achieving full objectiveness is still challenging for the field of cluster analysis. At least, the manuscript should describe a rigorous, proven workflow for users to set the parameter values and judge the analysis results, possibly involving iterations. Otherwise, the practical utility of the method in this manuscript may be severely limited because of the lack of rigor and reproducibility in the results.

2. Benchmarking and validation of the method. While the manuscript provides substantial testing of the method on a variety of systems, some aspects of benchmarking and validation is still insufficient.

(2.1) The definition of "classification performance", which use used extensively in this manuscript, is not fully explained. Is it aggregated for all clusters in the sample? How about individual values for different clusters? For a given cluster, are there any differences in false positives and negatives?

(2.2) The TOL case was used as a "more challenging" test case because the structures are "more similar to each other". Visually, though, the three different structures are still highly distinct. The practical challenge in this experimental test likely came more from higher noise/heterogeneity of these structures and their images instead of that they are "more similar to each other". The "TUD" with or without the flame would be a better benchmark for structures that are similar, but only simulation data is provided. The manuscript could benefit from more characterizations on how structural similarity affects classification performance (also associated with 2.3).

(2.3) In addition to structural similarity, the signal-to-noise ratio of the input images is another important factor affecting classification performance. Similar to how the manuscript has characterized the effect of localization uncertainty, the effect of signal-to-noise can be quantified, even with experimental data by subsampling the localization points.

3. The practical meaning of some demonstrations are either unclear or problematic.

(3.1) For the classifying of images from the same structure (Figure S9), there is no surprise that by setting K to 2, K-means can force the image into two groups. It is actually a perfect example of subjectiveness in K-means clustering. The division may reflect imperfections in the design of the distance metric and has no practical value. Moreover, rigorous statistics should have indicated the need to merge of these two classes. This example should be removed from the manuscript to avoid misleading the readers.

(3.2) For the classifying of images with continuous variabilities (Figure S7, S8 and S10), classification is actually a nice way to characterize the population, revealing its homogeneous and heterogeneous aspects, and facilitating particle averaging. However, this point is only superficially discussed in the manuscript. For example, a reader may wonder about the meaning of 2 vs 3 clusters for tetrahedral structures, while the actual value of this analysis is to reveal that the structural heterogeneity occurs at the top vertex but not the bottom three vertices (regardless the cluster number), and the "blur" in the unclassified averaging results is not from poor image resolution but the structural flexibility itself.

Reviewer #2 (Remarks to the Author):

In this manuscript Huijben et al present an approach for detecting structural heterogeneity in localization microscopy data - or rather, as it's more likely to be used, fitting localization microscopy data that has structural heterogeneity present. The work is well founded and unusual in that it takes a method (k-means clustering) which is simple and well established, making it easy to understand the likely failure modes.

Overall I think this work is well-evaluated on simple test structures and has notable advantages. I have two main comments:

1) The samples tested on are somewhat simple compared to likely biological targets, which are likely to show continuous distributions rather than discrete classes, possibly across multiple parameters. I think there should be some more consideration as to how performance may vary in less well-behaved samples.

2) It would be very interesting to see something other than a nuclear pore complex as a biological sample, though I recognise this may not be possible.

Reviewer #3 (Remarks to the Author):

In this manuscript, Huijben et al., presented a particle fusion method for detecting structural heterogeneity in single-molecule localization microscopy data. The proposed method demonstrated high classification accuracy even from the mixtures of four DNA origami structures, representing a step forward in the particle fusion field using SMLM data. The manuscript is well-organized and the authors' opinions on particle fusion are standout. Some concerns are listed as below and should be addressed before publication.

Major concerns:

1. The selection for number of clusters in the k-means clustering algorithm (page 8, 'K-means clustering' section in Methods part). The number of clusters, K , is an important parameter in k-means clustering. The small value of K will underdetermine the true clusters. In contrast, large value of K may generate many local structures.

- The authors mention that for the well-separated clusters, they will review the scatterplot of first three dimensions of multidimensional scaling (MDS) space. K can be chosen to match the number of clusters manually. Could the authors plot these scatterplots for different dataset? Are they looking similar?

- For the mirrored TUD-logo's dataset, the authors chose $K = 40$. However, the final results shown in the Fig. 3a-f only include two clusters. Could the authors explain why they have used such large value?

2. Ambiguous discussion in 2D particle fusion for NPC dataset. For gp210 dataset in Fig. 3g-l, the proposed method detected the structures forming ellipses. The authors suggested that the deformations were caused by the sample preparation. Due to the different locations of particles on the nuclear envelope, such ring like structure may form the ellipses when projected onto a 2D space. Therefore, both the projection process and the sample deformation can cause elliptical shapes. Could the authors discuss on how one could determine the original of such elliptical shape?

3. Clustering using eigen images (page 9 in Method part).

- For the large number of clusters, the authors further used hierarchical agglomerative clustering to reduce the number of clusters from K into C classes ($C < K$). Could the authors provide more details and, specifically, how many C classes they chose in each dataset?

- After hierarchical agglomerative clustering, how to get the final reconstructed images? What is the difference in result if one chooses to use C classes at the beginning, then follow k-mean clustering to get fusion images?

Some other minor concerns in the text:

1. Does misfolded data only appear in Fig. 2n-z? Or it is ubiquitous as long as we chose large number of classes. When the ground truth is absent, how to verify whether it is misfolded or not?

2. In Fig. 3 g-l, could the authors define the symbol 'e'?

3. In supplementary Fig. 7, how to define ellipticity? Is that the length ratio between minor axis and major axis?

4. In supplementary Fig. 7b, what does y-axis represent? Could the authors define the symbol 'S'?

5. In supplementary Fig. 8e, Could the authors define classification performance? Is that the ratio between the correctly assigned particles to the total particles?

RESPONSE TO REVIEWER COMMENTS

Reviewer #1 (Remarks to the Author):

This manuscript presents a method to classify single-molecule localization microscopy (SMLM) images of small structures. In a manner similar to particle classification and class-averaging in single-particle cryo-electron microscopy, this method can help SMLM resolving heterogenous subcellular structures to understand their structural organizations. Therefore, it is a useful tool to the community. The manuscript validated the method using simulated and experimental data. It is generally well written. The manuscript can be accepted if the comments below are satisfactorily addressed:

1. Clustering method. Any classification method has two components: a distance metric telling the difference between two particle images, and a cluster analysis method. The distance metric in this manuscript is mostly based on what the authors previously established for single particle image alignment, and the manuscript used the centroid-based K-means method for cluster analysis. While K-means is widely used and has the best performance when the clusters are more or less spherical, there are many other cluster analysis methods available, some of them avoiding the limitation of K-means that a pre-defined cluster number is needed. It is therefore important for the manuscript to:

(1.1) Provide a rationale of why K-mean method is chosen (even a very simple one) and a brief

discussion of other cluster analysis methods potentially suitable for this task, as well as their advantages and drawbacks.

Response: We acknowledge the reviewer's concerns regarding the choice of k-means as our clustering approach. The reason why we opt for k-means is three-fold. First of all, k-means is the most simple, well-known and best studied algorithm for clustering. Secondly, the k-means algorithm has only one free parameter (K), and this parameter is rather intuitive on the MDS maps compared to the free parameter(s) in other clustering approach. Mean-shift clustering method, as an example, can potentially be used as an alternative to k-means, however, the bandwidth size selection itself is challenging and in our case does not have a physical meaning as the distance between the points in MDS maps is based on the dissimilarity values. Other complex clustering approaches like DBSCAN or OPTICS have similar difficulties in tuning even more free parameters which is prohibitive for non-expert end-users. Thirdly, visual inspection of the MDS scatter plots shows that clusters appear to be spherical, which is a requirement for using k-means clustering.

We agree with the reviewer that the rationale for using k-means over all other possible clustering methods is crucial, and we therefore added a new paragraph to the main text devoted to the reasoning of k-means (Line 17, page 2 of main text).

(1.2) Establish an objective way of determining the cluster number, K , and an objective metric to test the validity and quality of cluster analysis results, especially for samples without sufficient prior knowledge. Currently, the determination of K is rather arbitrary and highly subjective. For example, in the case of analyzing the TOL origami images, it is already known that three types of structures exist in the data. However, setting $K = 3$ resulted in seemingly acceptable clustering but completely missed the L class. L was observed only when increasing K to 5 (not even 4). In other words, the L class will not be discovered unless its existence is already known and parameters are tweaked specifically for its appearance. This is a huge limitation to practical applications in making biological discoveries. Similarly, the manuscript states that a large K needs to be used for the purpose of discovering rare populations, which is another case where user choices affect analysis outcome. It is understood that achieving full objectiveness is still challenging for the field of cluster analysis. At least, the manuscript should describe a rigorous, proven workflow for users to set the parameter values and judge the analysis results, possibly involving iterations. Otherwise, the practical utility of the method in this manuscript may be severely limited because of the lack of rigor and reproducibility in the results.

Response: As for every clustering approach, the choice for the optimal number of clusters is always tricky. We agree with the reviewer that we did not pay enough attention to our logic for the choice of K for each dataset. Next to the use of prior knowledge about the dataset, we have added two measures that together can guide the user in finding the optimal value for K : the scatter plots of the first three dimensions of the MDS coordinates (**Suppl. Fig. 1**) and silhouette values for multiple values for K as cluster evaluation metric (**Suppl. Fig 2**). Both plots are shown for the most important datasets of the paper: the digits, the letters, the mirrored TUD structures and the nuclear pores.

Taking the letters (TOL) dataset as example, both the MDS scatter plots as well as the silhouette values indicate that $K=3$ is the optimal choice. Indeed, classifying the particles into three classes gives correct classification, both when the structures are imaged separately (**Suppl. Fig. 6d**) or together (**Suppl. Fig. 7a-c**). However, the presence of many misfolded structures prevents a nice reconstruction of the simplest structure, the letter "L". When classifying the particles into respectively 4 and 5 classes, the misfolds are captured in separate classes and the "L"-structure is nicely reconstructed.

We made the following changes to make the choice of K clearer to the reader:

- We added two new supplementary figures (**Suppl. Fig. 1** and **Suppl. Fig. 2**) containing the MDS scatter plots and silhouette plots for multiple datasets, including extensive captions explaining the choice of K .
- The calculation of the silhouette measure is added to Methods (Line 20, page 9).
- We added a paragraph to the main text discussing the guidelines for choosing the optimal value for K (Line 26, page 2).
- We added additional sentences to the main text to make clear for every dataset how the value of K is determined.

2. Benchmarking and validation of the method. While the manuscript provides substantial testing of the method on a variety of systems, some aspects of benchmarking and validation is still insufficient.

To benchmark and validate our method, we were motivated by a recent upload on bioRxiv (Sabinina *et al.*). In general, there are two techniques commonly used to cluster based on pairwise dissimilarities: clustering based on a spatial embedding of the particles (for example MDS, like in our approach) and direct hierarchical clustering (HAC, like done by Sabinina *et al.*). We have implemented and optimized both strategies and tested them on multiple datasets.

We have added a Supplementary Note (**Supplementary Note 1**) and supplementary figure (**Supplementary Fig. 12**) in which we compare our method to the direct hierarchical clustering approach and show that our method performs significantly better than the hierarchical clustering approach.

(2.1) The definition of "classification performance", which is used extensively in this manuscript, is not fully explained. Is it aggregated for all clusters in the sample? How about individual values for different clusters? For a given cluster, are there any differences in false positives and negatives?

Response: We define the classification performance as the percentage of correctly classified particles. To make this clear for the reader, we have added this definition to the main text (Line 45, page 2 of main text).

This definition of the classification performance is indeed a metric in which all clusters are aggregated. A valid point is that this value does not give information about the different clusters. We therefore show the confusion matrices of the classification, which is a table that indicates how many particles of each class are classified as a certain class. This confusion matrix allows for investigation of all false/true positive/negative combination between the different classes and can only be constructed when the ground truth class label of each particle is available. The confusion matrices are already given in the manuscript in **Fig. 2** and **Suppl. Fig. 6** for the classifications where the ground truth is known. For completion, we have added the matrices to **Suppl. Fig. 8**.

(2.2) The TOL case was used as a "more challenging" test case because the structures are "more similar to each other". Visually, though, the three different structures are still highly distinct. The practical challenge in this experimental test likely came more from higher noise/heterogeneity of these structures and their images instead of that they are "more similar to each other". The "TUD" with or without the flame would be a better benchmark for structures that are similar, but only simulation data is provided. The manuscript could benefit from more characterizations on how structural similarity affects classification performance (also associated with 2.3).

Response: The reviewer correctly indicates that we incorrectly state that the TOL structures are “more similar to each other”. The reason why we consider the TOL data more challenging is three-fold. First of all, the structures in the TOL dataset only contain of 3-4 dots. This makes the distinction between the letters harder especially when one or more dots are missing due to low labelling density. Secondly, the quality of different structures in the TOL dataset differs per class. The localization uncertainty of the localizations in the letter “L” is approximately three times worse than for the letters “T” and “O” (**Suppl. Fig. 7m**). Therefore, it is harder to classify this letter correctly. Thirdly, recognizing misfolded DNA-origamis is more challenging due to their similarity to the underlabelled “TOL” structures (3-4 dots). In contrast, the TUD logo has a well-defined structure, which makes the particle picking process (misfold/correctly folded) easier.

To clarify this point to the readers, we added an extra sentence in the main text (Line 9, page 3 of main text), added a new panel to **Suppl. Fig 6** showing the high degree of similarity between the different TOL structures, and elaborated the explanation in the caption of **Suppl. Fig. 7**.

(2.3) In addition to structural similarity, the signal-to-noise ratio of the input images is another important factor affecting classification performance. Similar to how the manuscript has characterized the effect of localization uncertainty, the effect of signal-to-noise can be quantified, even with experimental data by subsampling the localization points.

Response: Considering the image formation model of single-molecule localization microscopy (SMLM), there are two factors which affect the quality of the acquired images in case of PAINT imaging: localization uncertainty due to the limited photon count and underlabelling due to incomplete fluorescent labelling of the target structures. In that sense, SNR and the conventional Gaussian or Poissonian noise model is not applicable to SMLM localization data (list of coordinates) as it is for other microscopy techniques. We have studied both of the mentioned sources of data degradation in **Suppl. Fig. 10e**.

3. The practical meaning of some demonstrations are either unclear or problematic.

(3.1) For the classifying of images from the same structure (Figure S9), there is no surprise that by setting K to 2, K-means can force the image into two groups. It is actually a perfect example of subjectiveness in K-means clustering. The division may reflect imperfections in the design of the distance metric and has no practical value. Moreover, rigorous statistics should have indicated the need to merge of these two classes. This example should be removed from the manuscript to avoid misleading the readers.

Response: In SMLM, although the imaged particles (in one class) are similar, they are not exactly the same due to the stochasticity of the localization generation in each particle in addition to underlabelling. With this example on the classification of one-class data, our aim was to show that our approach can potentially be used to study the binding site affinity distribution and fluorescent label incorporation in a SMLM experiment. However, we agree with the reviewer that this example could mislead the readers in understanding classification of homogeneous data and therefore we removed it in the revised manuscript.

(3.2) For the classifying of images with continuous variabilities (Figure S7, S8 and S10), classification is actually a nice way to characterize the population, revealing its homogeneous and heterogeneous

aspects, and facilitating particle averaging. However, this point is only superficially discussed in the manuscript. For example, a reader may wonder about the meaning of 2 vs 3 clusters for tetrahedral structures, while the actual value of this analysis is to reveal that the structural heterogeneity occurs at the top vertex but not the bottom three vertices (regardless the cluster number), and the “blur” in the unclassified averaging results is not from poor image resolution but the structural flexibility itself.

Response: We agree with the reviewer here and added a sentence to the main text (Line 14, page 4 of main text) to clarify the purpose of this experiment and emphasized on our finding that we were able to capture the structural variation of the tetrahedron nanostructures using our developed classification pipeline.

Reviewer #2 (Remarks to the Author):

In this manuscript Huijben et al present an approach for detecting structural heterogeneity in localization microscopy data - or rather, as it's more likely to be used, fitting localization microscopy data that has structural heterogeneity present. The work is well founded and unusual in that it takes a method (k-means clustering) which is simple and well established, making it easy to understand the likely failure modes.

Overall I think this work is well-evaluated on simple test structures and has notable advantages. I have two main comments:

1) The samples tested on are somewhat simple compared to likely biological targets, which are likely to show continuous distributions rather than discrete classes, possibly across multiple parameters. I think there should be some more consideration as to how performance may vary in less well-behaved samples.

Response: We have applied our algorithm to several experimental DNA-origami nanostructures with different realistic imaging conditions (photon count, labelling density and structure size) to characterize the pros and cons of our developed pipeline. For biologically relevant structures, we were unfortunately limited to the currently available data in SMLM community, i.e. NPCs. On that, however, we showed how the classification algorithm can separate particles based on a continuous feature of the underlying structure (ellipticity of the pores). We believe that this is promising in the applicability of our approach in the study of other possible structures in the future. More importantly, most biological structures studied with SMLM in combination with particle averaging typically consist of a number of distinct binding sites (perfect example, the nuclear pore complex). Next to the possible continuous shape variations of the system, the addition or deletion of certain domains/subgroups of the structure will result in discrete classes, which is a common phenomenon in biology.

2) It would be very interesting to see something other than a nuclear pore complex as a biological sample, though I recognise this may not be possible.

Response: Unfortunately, there are not many repeated biologically relevant structures publicly available at this time other than NPCs and DNA-origami nanostructures.

Reviewer #3 (Remarks to the Author):

In this manuscript, Huijben et al., presented a particle fusion method for detecting structural

heterogeneity in single-molecule localization microscopy data. The proposed method demonstrated high classification accuracy even from the mixtures of four DNA origami structures, representing a step forward in the particle fusion field using SMLM data. The manuscript is well-organized and the authors' opinions on particle fusion are stand-out. Some concerns are listed as below and should be addressed before publication.

Major concerns:

1. The selection for number of clusters in the k-means clustering algorithm (page 8, 'K-means clustering' section in Methods part). The number of clusters, K , is an important parameter in k-means clustering. The small value of K will underdetermine the true clusters. In contrast, large value of K may generate many local structures.

- The authors mention that for the well-separated clusters, they will review the scatterplot of first three dimensions of multidimensional scaling (MDS) space. K can be chosen to match the number of clusters manually. Could the authors plot these scatterplots for different dataset? Are they looking similar?

Response: The reviewer makes a valid point about the missing scatter plots. The choice for the parameter K is deduced from prior knowledge in combination with the scatter plot of the first three dimensions of the MDS space. To show that these scatter plots give a clear indication for the choice of K , we have added a new supplementary figure (**Suppl. Fig. 1**), where we show the MDS scatter plots of four different datasets, of experimental and simulated data, of datasets that contain discrete and continuous variation.

The added supplementary figure, together with the caption, illustrate that when the dataset contains discrete variations, the MDS scatter plot shows distinct clusters that indicate to the user which value of K best suits the classification. In case of a continuous variation, the MDS scatter plot will form a single cluster in which the images are sorted based on similarity. Here, any value of K will suffice for classification, depending on the preference of the user.

- For the mirrored TUD-logo's dataset, the authors chose $K = 40$. However, the final results shown in the Fig. 3a-f only include two clusters. Could the authors explain why they have used such large value?

Response: Indeed, for 50% DOL data, we used $K = 40$. The reason is that the MDS scatter plot does not show two clear groups and the silhouette graph is flat. Both observations do not indicate an optimal K , so we use the general advice to go for a large K . In order to correctly classify the few mirrored particles, the strategy is to cluster the MDS space into many clusters, and group them further using the optional eigen image approach. Since the number of obtained K clusters is higher than the desired C classes, the clusters are grouped further using the eigen-image method with $C = 2$, which is described in the method section of the manuscript.

2. Ambiguous discussion in 2D particle fusion for NPC dataset. For gp210 dataset in Fig. 3g-l, the proposed method detected the structures forming ellipses. The authors suggested that the deformations were caused by the sample preparation. Due to the different locations of particles on the nuclear envelope, such ring like structure may form the ellipses when projected onto a 2D space. Therefore, both the projection process and the sample deformation can cause elliptical shapes. Could the authors discuss on how one could determine the original of such elliptical shape?

Response: We thank the reviewer for this comment. The reviewer is correct and the resulting ellipticity could have originated from the projection of tilted circles. However, a circle needs to have an inclination of 25.8° with respect to the imaging plane, to create an elliptical projection with an ellipticity of 0.9. We consider this angle way too big to happen in experiments, which is the reason we

assume the aligned ellipses are caused by the sample preparation. We added this consideration to the main text (Line 50, page 3 of main text).

3. Clustering using eigen images (page 9 in Method part).

- For the large number of clusters, the authors further used hierarchical agglomerative clustering to reduce the number of clusters from K into C classes ($C < K$). Could the authors provide more details and, specifically, how many C classes they chose in each dataset?

Response: We agree with the reviewer that the main text and figure captions do not contain sufficient information about the specific values for K and C , when the further clustering with the eigen image approach is used. In essence, the eigen image approach is an addition to the default pipeline and is only used in cases when there is a large imbalance in the number of particles per cluster, namely: the mirrored TUD logos (**Fig. 3a-f**) and the 9-fold symmetric nuclear pores (**Suppl. Fig. 8**). For all other results, the standard pipeline (**Fig. 1**) is used, and only the value for K is relevant.

To make the details of the use of the eigen image approach clear to the reader, we have made the following changes:

- For the mirrored TUD structures, we have added the values of K and C to the figure caption of **Fig. 3a-f**.
- For the 9-fold symmetric nuclear pores, we have added the value of C to the figure caption of **Suppl. Fig. 8**.
- For both cases, we have added the values of C , and the definition of this value, to the Methods section (Line 16, page 9 of Methods).

- After hierarchical agglomerative clustering, how to get the final reconstructed images?

Response: The images belonging to each class C ($C < K$) are reconstructed together using the particle fusion pipeline (Heydari *et al.* Nature Methods 2018). We have added this explanation to the Methods section about the eigen image approach (Line 17, page 9 of Methods).

What is the difference in result if one chooses to use C classes at the beginning, then follow k-mean clustering to get fusion images?

Response: If the dataset contains two clearly separated classes of structures, both methods will give the same result. Because the particles in the MDS space will be clearly separated into 2 clusters, and clustering into K or C groups will both be fine. However, the two approaches will give different results when one of the classes contains significantly less particles than the other. In this case, k-means clustering of the MDS space will not result in correct classification, since k-means tends to find spherical clusters and will not separate the small class from the big class (when they are close together, which is usually the case when the two classes are structurally similar). An example is given in **Suppl. Fig. 1g-l**, where the mirrored TUD structures (orange dots) are separated from the normal particles (blue dots), but k-means with $K = 4$ is needed to correctly cluster the two groups. In this example, it is necessary to cluster with a higher K (to capture the small cluster) and further group the clusters with the eigen image approach with a lower C .

Some other minor concerns in the text:

1. Does misfolded data only appear in Fig. 2n-z? Or it is ubiquitous as long as we chose large number of classes. When the ground truth is absent, how to verify whether it is misfolded or not?

Response: All DNA origami structures contain misfolds, the point is whether the particle picking process can distinguish the misfolded structures from the correctly folded structures and remove them from the dataset. Well-defined structures, like the TUD logo, have a distinct appearance and the particle picking process can detect the misfolded structures, that is the reason why the TUD dataset does not suffer from a noticeable number of misfolded structures. The structures of the letters dataset, on the contrary, only consist of a few dots. This makes it hard for the particle picking process to distinguish a misfolded structure from a correctly folded one, especially in combination with underlabelling. We have added this reason for the appearance of misfolds to the main text (Line 16, page 3 of main text).

When the ground truth is absent, we indeed cannot verify with certainty whether particles are misfolded or not. Although, visual inspection of single particles gives a clear indication that the particle does not resemble one of the known designs (see **Suppl. Fig. 6h**).

2. In Fig. 3 g-l, could the authors define the symbol 'e'?

Response: The symbol 'e' in **Fig. 3g-l** defines the ellipticity, which we define as the ratio of the length of the major axis over the length of the minor axis of the ellipse that is fitted to the data.

We have added the definition of the ellipticity to the caption of **Fig. 3** of the main text and a detailed explanation of how we fit the ellipses to the Methods section of the main text (Line 29, page 9 of main text).

3. In supplementary Fig. 7, how to define ellipticity? Is that the length ratio between minor axis and major axis?

Response: See our reply to the previous question for our explanation of the definition and calculation of the ellipticity. For clarity, we have added this information the Methods section of the main text (Line 29, page 9 of the main text).

4. In supplementary Fig. 7b, what does y-axis represent? Could the authors define the symbol 'S'?

Response: **Suppl. Fig. 9b** (was originally 7b, but is now renumbered) shows the polar histograms of the orientation of the elliptical fits for all particles, the particles classified as 'circle' and the particles classified as 'ellipse'. The y-axis in these plots indicates the count, e.g. the number of particles of which the elliptical fit gives this particular orientation. For clarity, we have added a y-label to the polar histograms in **Suppl. Fig. 9b**.

We thank the reviewer for noticing that we forgot to give the definition of the S parameter mentioned in **Suppl. Fig. 9b**. The S parameter represents the *order parameter*, which is defined as $\langle \cos(2(\theta - \bar{\theta})) \rangle$, where θ represents the orientation of the ellipse, $\bar{\theta}$ the director which is the average angle of all ellipses, and $\langle \dots \rangle$ the population average over all ellipses of the respective class. By definition, a value for the order parameter of 1 indicates complete alignment of all ellipses and a value of 0 indicates a homogeneously distributed random orientation. The order parameter metric is calculated to show that the class of ellipses are mutually more aligned than the particles that are classified as circles.

We have added definition of the order parameter to the Methods section of the main text (Line 36, page 9 of main text).

5. In supplementary Fig. 8e, Could the authors define classification performance? Is that the ratio between the correctly assigned particles to the total particles?

Response: The reviewer correctly indicates that we did not clearly state our definition of the classification performance. See our reply to question 2.1 of the first reviewer for our explanation and how we have added this topic to the manuscript and supplement.

REVIEWERS' COMMENTS

Reviewer #1 (Remarks to the Author):

The revised manuscript has satisfactorily addressed the concerns previously raised by this reviewer.

Reviewer #2 (Remarks to the Author):

My concerns have been addressed and the manuscript is now suitable for publication

Reviewer #3 (Remarks to the Author):

The authors revised the manuscript and addressed my previous concerns. I recommend publication of this manuscript.